# Proteomic Profiling Identifies Specific Leukemic Stem Cell-Associated Protein Expression Patterns in Pediatric AML Patients

**DOI:** 10.3390/cancers14153567

**Published:** 2022-07-22

**Authors:** Marianne Agerlund Petersen, Carina Agerbo Rosenberg, Marie Bill, Marie Beck Enemark, Ole Rahbek, Anne Stidsholt Roug, Henrik Hasle, Bent Honoré, Maja Ludvigsen

**Affiliations:** 1Pediatrics and Adolescent Medicine, Aarhus University Hospital, 8200 Aarhus N, Denmark; hasle@dadlnet.dk; 2Department of Hematology, Aarhus University Hospital, 8200 Aarhus N, Denmark; carose@rm.dk (C.A.R.); marie.bill@rm.dk (M.B.); mariem@rm.dk (M.B.E.); annrou@rm.dk (A.S.R.); majlud@rm.dk (M.L.); 3Department of Clinical Medicine, Aarhus University, 8200 Aarhus N, Denmark; 4Department of Orthopedic Surgery, Aalborg University Hospital, 9000 Aalborg, Denmark; o.rahbek@rn.dk; 5Department of Clinical medicine, Aalborg University, 9000 Aalborg, Denmark; 6Department of Biomedicine, Aarhus University, 8000 Aarhus C, Denmark; bh@biomed.dk

**Keywords:** mass spectrometry, proteomics, pediatric acute myeloid leukemia, hematopoietic stem cells

## Abstract

**Simple Summary:**

Acute myeloid leukemia is an aggressive cancer in children and novel therapeutic tools are warranted to improve outcomes and reduce late effects in these patients. In this study, we isolate and explore the protein profiles of leukemic stem cells and normal hematopoietic stem cells from hematologically healthy children. Differences in protein profiles between leukemic and normal hematopoietic stem cells were identified. These results provide an insight into the disrupted biological pathways in childhood acute myeloid leukemia. Moreover, differences in protein profiles may serve as potential targets for future therapies specifically aiming at the disease-propagating leukemic stem cells while omitting the normal hematopoietic stem cells.

**Abstract:**

Novel therapeutic tools are warranted to improve outcomes for children with acute myeloid leukemia (AML). Differences in the proteome of leukemic blasts and stem cells (AML-SCs) in AML compared with normal hematopoietic stem cells (HSCs) may facilitate the identification of potential targets for future treatment strategies. In this explorative study, we used mass spectrometry to compare the proteome of AML-SCs and CLEC12A+ blasts from five pediatric AML patients with HSCs and hematopoietic progenitor cells from hematologically healthy, age-matched controls. A total of 456 shared proteins were identified in both leukemic and control samples. Varying protein expression profiles were observed in AML-SCs and leukemic blasts, none having any overall resemblance to healthy counterpart cell populations. Thirty-four proteins were differentially expressed between AML-SCs and HSCs, including the upregulation of HSPE1, SRSF1, and NUP210, and the enrichment of proteins suggestive of protein synthesis perturbations through the downregulation of EIF2 signaling was found. Among others, NUP210 and calreticulin were upregulated in CLEC12A+ blasts compared with HSCs. In conclusion, the observed differences in protein expression between pediatric patients with AML and pediatric controls, in particular when comparing stem cell subsets, encourages the extended exploration of leukemia and AML-SC-specific biomarkers of potential relevance in the development of future therapeutic options in pediatric AML.

## 1. Introduction

Conventional chemotherapy remains the mainstay in the treatment of acute myeloid leukemia (AML) in children [1,2]. Over the past three decades, the outcome for children with AML has improved considerably, and overall survival rates have risen to around 70% [3]. However, relapse occurs in up to 30% of the patients and is often associated with a dismal outcome [4]. Furthermore, cumulating evidence has shown that the necessary use of intensive chemotherapy to obtain complete remission both at diagnosis and relapse is associated with severe late effects [2,5], which in childhood cancer survivors may last a lifetime. Thus, much leukemia research is centered around the identification of leukemia-specific targets to reduce “on-target/off-tumor” toxicities. In AML, this quest is continuously ongoing, and, so far, the challenge has been to identify targets that are present on malignant cells including leukemia-initiating stem cells (AML-SCs) but absent on healthy hematopoietic stem cells (HSCs) [6,7,8,9,10]. In both adult and pediatric AML, novel targeted anticancer agents, such as small-molecule inhibitors and antibody-based and cellular immunotherapies directed at, e.g., CD33 and CD123, are being investigated [11,12,13,14] (clinicaltrials.gov identifier: NCT03672539, NCT03971799, NCT04158739). However, although frequently expressed in AML, these antigens are not entirely leukemia-specific [7,8,10,15,16]. Other antigen-directed treatments are in the pipeline, and one intriguing target is the C-type lectin domain family 12 member A receptor, CLEC12A, also known as CLL-1. This antigen has been extensively studied in adult AML, and is often expressed on both leukemic blasts and leukemic stem cells [7,8,17,18,19] but is notably absent on HSCs [9,17,20]. Hence, CLEC12A-directed therapy is a budding treatment option in the AML armamentarium [21].

Acute myeloid leukemia is a highly heterogeneous disease at both immunophenotypic and genotypic levels [9,22]. Some genetic aberrations are exclusively detected in pediatric AML, whereas others are confined to adults [23,24]. As AML is a rare childhood cancer [25], much insight into AML biology is derived from adult studies and is therefore not inherently generalizable to pediatric patients. Moreover, age-related biological variations have been described in healthy human hematopoiesis, with functional and transcriptional differences being found between young and elderly HSCs [26,27]. Hence, explanatory models used to facilitate the understanding of pediatric AML biology are in some areas based on knowledge from adult AML research. This relative paucity of knowledge in pediatric AML biology extends to the field of proteomics. Proteins are often the endpoint of targeted therapies in modern precision medicine, and putative disease-specific biomarkers using a proteomic approach have previously been identified in adult AML [6,28,29], but are far less explored in pediatric patients.

In many CD34 positive AML subtypes, the AML-SC population residing within the CD34+CD38− compartment is conceived as the cellular foundation for leukemogenesis [30,31,32]. In the case of CD34 negative AML, however, the leukemia-initiating cells are often contained within other immunophenotypic cell compartments, and in these AML subtypes, residual HSC may comprise the CD34+CD38− subset [33,34,35]. The AML-SC compartment often constitutes a minute fraction of the bulk leukemia blast cells [36,37]. Correspondingly, the healthy immature HSCs and hematopoietic progenitor cells (HPCs), including granulocyte–macrophage progenitor (GMP) cells, common myeloid progenitor (CMP) cells, and megakaryocyte–erythroid progenitor (MEP) cells, are also rare populations within the cellular landscape in the bone marrow (BM) [38]. Consequently, the isolation of rare immunophenotypically well-defined cell populations such as HSCs, HPCs, and AML-SCs by fluorescence-activated cell sorting (FACS) is a prerequisite in order to perform systematic characterization of these biologically relevant subsets. In the present study, we explored protein profiles in purified CD34+CD38− AML-SCs, CD34+CD38+CLEC12A+, and CD34−CD38+CLEC12A+ leukemic blasts in BM samples from five pediatric AML patients using liquid chromatography–tandem mass spectrometry (LC-MS/MS). We describe unique proteins selectively dysregulated in AML-SCs and blasts from pediatric AML patients compared with purified HSC and HPC populations from five age-matched controls. Lastly, we explored pathway perturbations based on the differential protein expression patterns that may potentially be targetable in future treatment schemes when the data have been further validated in larger independent pediatric AML cohorts.

## 2. Materials and Methods

### 2.1. Patients and Samples

Cryopreserved mononuclear cells (MNCs) from diagnostic BM samples from five pediatric patients with AML were included in the study based on the availability of sufficient pretherapeutic material (Supplementary methods). The samples from the leukemic patients were collected as part of the routine diagnostic workup. Clinical data were obtained from the AML database administered by the Nordic Society of Pediatric Oncology and Hematology (NOPHO) (Table 1). Flow cytometry data from the five pediatric AML patients, and the mutational status of the AML-SC compartment in three of the patient samples were previously determined [39]. In addition, cryopreserved MNCs from five hematologically healthy pediatric patients (aged 8 to 12 years, three females and two males) were included as controls. Bone marrow samples were collected from the controls after informed consent from the parents and during planned orthopedic surgical procedures involving the pelvic/hip region.

### 2.2. Fluorescence-Activated Cell Sorting

Cells were sorted by FACS using a BD FACSAria III (BD Biosciences, San Jose, CA, USA) to obtain proteomic profiles on isolated cell populations as described in detail in Appendix A. In brief, cryopreserved MNCs were thawed, stained (reagents listed in Appendix A), and sorted into proteomic lysis buffer. From the AML patients, the following five subsets were sorted: AML-SC (CD34+CD38−), progenitor cells 1 (PC1; CD34+CD38+CLEC12A+), PC2 (CD34+CD38+CLEC12A−), blast cells 1 (BC1; CD34–CD38+CLEC12A+), and BC2 (CD34−CD38+CLEC12A−) (Appendix A). Notably, these terminologies were selected for reasons of simplicity; hence, leukemic blasts were present in both the PC and BC compartments. From the healthy controls, the following seven subsets were sorted: HSC (Lin−CD34+CD38−CD90+/−CD45RA−), CMP1 (Lin−CD34+CD38+CD90−CD45RA−CD123+CLEC12A+), CMP2 (Lin−CD34+CD38+CD90−CD45RA−CD123+CLEC12A–), GMP1 (Lin−CD34+CD38+CD90−CD45RA+CD123+CLEC12A+), GMP2 (Lin−CD34+CD38+CD90−CD45RA+CD123+CLEC12A−), MEP (Lin−CD34+CD38+CD90−CD45RA−CD123−), and CLEC12A+ Precursors (C12A+ Pre; Lin–CD34−CD38+CLEC12A+) (Appendix A). The CMP1/2, GMP1/2, MEP, and C12A+ Pre were collectively named HPCs. The fraction of the various cell subsets is shown in Appendix A and the purity analyses are provided in Appendix A.

### 2.3. Proteomics

Proteomic profiling of the purified cell populations was performed by label-free quantification nano LC-MS/MS with the purpose of identifying differentially expressed proteins, concentrating on differences between the leukemic AML-SC, PC1, and BC1 samples and the HSCs from the controls. The procedure is described in detail in the Appendix A. Sample preparation was conducted with the in-stage Tip (iST) method [40]. Mass spectrometry analysis was performed as previously described using an Orbitrap Fusion Tribrid instrument (Thermo Fisher Scientific Instruments, Waltham, MA, USA) [41]. Peptide and protein identifications were performed by searching raw data against the Homo sapiens database downloaded 9 February 2020 from UniProt (www.uniprot.org (accessed on 9 February 2020)). After filtering, a total of 456 proteins were identified in 49 eligible samples (Appendix A, Appendix A) and used in principal component analyses. Because the analyses were focused on exploring the proteome of the AML-SC and the CLEC12A+ blasts, the CLEC12A− PC2 and BC2 subsets were excluded in downstream analyses.

### 2.4. Statistical Analysis

Median and range were reported for the cell population frequencies unless otherwise stated. Student’s t-test was used for comparison of expression levels in the cell populations to identify significantly differentially expressed proteins (*p* < 0.05). Graph Pad Prism 9.4.3 (GraphPad Software, San Diego, CA, USA) was used for creating graphs. Principal component analysis (PCA) was performed in RStudio (RStudio: integrated development environment for R, Version 1.3.1093, Boston, MA, USA). Since the present analysis was a proteomic discovery-based study, we abstained from performing correction for multiple hypothesis testing since although this provides a wanted decrease in type 1 errors, it also increases type 2 errors with the risk of overlooking putative markers.

## 3. Results

### 3.1. Protein Profiles of Leukemic Cell Subsets from Pediatric AML Are Partly Patient Specific and Partly Cell Subset Specific

Among the five AML patients were one male and four females between 2 and 12 years of age (Table 1). Three patients suffered from induction death, one patient died after relapse, and one patient died due to resistant disease. In pAML29, the leukemia cells harbored the inversion of chromosome 16 (inv(16)(p13q23)) together with a FLT3 tyrosine kinase domain (TKD) mutation. Two patients (pAML17 and pAML21) had KMT2A gene rearrangement AML (t(9;11)(p22;q23)). The last two cases (pAML20 and pAML23) had a normal karyotype and no known molecular genetic aberrations. Results from cytogenetic and molecular genetic analyses on the bulk leukemic cells and on FACS-sorted CD34+CD38− stem cells from diagnostic BM material are provided in Table 1. From the routine flow cytometry analyses, at diagnosis, the BM leukemic blast fractions were 84–91%. Two of the patients (pAML17 and pAML21) had CD34 negative leukemia, whereas in pAML23 and pAML29, the blasts displayed a broad CD34 expression and were termed CD34 positive. Although the pAML20 leukemia was categorized as CD34 positive (>5% CD34 positive AML cells), only 8.6% of the leukemic blasts were CD34 positive (Table 1).

Initially, by investigating the protein expression patterns of the patient samples, we observed a separation in principal component 1 of the pAML17, pAML20, and pAML21 samples from the CD34 positive pAML23 and pAML29 samples, suggestive of patient-dependent protein expression patterns (Figure 1). In addition, two BC1 clusters were observed: one cluster comprising pAML17 and pAML21, together with pAML20 samples; another, pAML23 and pAML29 samples, indicative of two distinguishable protein expression patterns. Focusing on the AML-SCs, there was no clear clustering of all AML-SC samples from the leukemic blast subsets in principal components 1 and 2. However, in principal component 3, some separation was apparent, suggesting that the differences in protein expression between AML-SCs and blasts may also be partly dependent on the specific cell subset (Figure 1).

### 3.2. The AML-SC Protein Profile Differs in CD34 Negative and CD34 Positive AML compared with HSCs and HPCs

Next, we compared the protein profile of the AML-SC populations with the HSCs and HPCs from the controls. Clearly, the AML-SCs from pAML23 and pAML29 were distributed differently from all other samples, while the AML-SCs from pAML17, pAML20, and pAML21 were situated parallel to the HSCs and HPCs (Figure 2a). As specified in Table 1, cytogenetic analyses of the purified AML-SCs from pAML17 and pAML21 were negative for the KMT2A rearrangement by FISH, otherwise characterizing the leukemia [39]. Interestingly, in the PCA, the AML-SCs from these two particular cases were mapped near the HSCs, whereas the AML-SCs from pAML20, pAML23, and pAML29 were separate from this healthy counterpart. Thus, these results indicate that the AML-SC proteome in CD34 negative AML shares some resemblance with HSCs/HPCs, whereas AML-SCs in CD34 positive AML are largely dissimilar to their healthy counterparts.

After chemotherapy, HSCs are fundamental to the restoring of BM function. We hypothesized that some proteins may affect alternative pathways of potential importance to the AML-SCs but not to HSCs, and such proteins or pathways might represent potential therapeutic targets. To this end, we compared the protein expression profile of AML-SCs to HSCs. This analysis revealed 34 differentially expressed proteins, of which 16 (47%) were expressed at lower levels in the AML-SCs and 18 (53%) were present at higher levels (Figure 2b, Appendix A), the latter including 10 kDA heat shock protein, mitochondrial (HSPE1), serine/arginine-rich splicing factor 1 (SRSF1), and nucleopore membrane glycoprotein NUP210 (Appendix A). An ingenuity pathway analysis (IPA) [42] of the 34 differentially expressed proteins indicated significant downregulation of the pathway “signaling through EIF2” (eukaryotic initiation factor 2), a pathway essential to protein synthesis by the initiation of translation. Interestingly, the changes in the proteins RPL13, RPL14, RPL24, RPLP2, and RPS8, participating in this pathway, could be explained by the decreased activity of their common transcription factor, MLXIPL (Figure 2c).

### 3.3. The Blast Proteome Resembles Healthy Progenitors in CD34 Negative AML but Is Unique in CD34 Positive AML

Similar to the AML-SCs from pAML23 and pAML29, we observed that the leukemic blasts from these two samples mapped together and at a different site than the remaining three patient samples and the HSCs and HPCs (Figure 3a). In contrast, leukemic blasts from the CD34 negative pAML17 and pAML21 together with pAML20, mapped alongside HPCs and, in particular, near the GMP1s and CLEC12A+ Pre (Figure 3a), which is in accordance with gene expression profiling data in adult CD34 negative AML [35]. We observed a tendency of the BC1 subsets from pAML17, pAML20, and pAML21 to cluster, whereas the PC1 subsets from these three samples were more spread out. Collectively, the leukemic blasts in two CD34 positive AML samples showed a tendency toward a unique and largely different-from-normal protein profile, whereas the protein profile of the leukemic blasts in pAML17, pAML20, and pAML21 seemed to be more similar to the healthy progenitors.

Next, to explore how protein expression patterns specifically distinguished PC1 and BC1 from HSCs, the following analyses were performed: First, when comparing PC1 to HSCs, 48 (77%) proteins were identified as upregulated and 14 (23%) proteins downregulated (Figure 3b and Appendix A). Next, a substantial difference in protein expression was observed in the BC1 vs. HSC analysis, with 132 proteins being identified including 103 (78%) upregulated and 29 (22%) downregulated proteins (Figure 3c and Appendix A). Among the upregulated proteins in both analyses, NUP210 was identified together with calreticulin (CALR).

In summary, 17 proteins were differently expressed in the AML-SCs, PC1, and BC1 in the AML samples compared with the HSCs from the controls (Appendix A). Of these 17 proteins, 11 (65%) were upregulated and 6 (35%) were downregulated. Moreover, 10, 9, and 76 were unique in the AML-SCs (Appendix A), PC1, and BC1, respectively, compared with the HSCs (Appendix A). The 17 shared proteins might represent targets in the development of novel agents aiming at both AML-SCs and blasts in children with AML. In theory, such agents would be likely to spare residual healthy HSCs in the patients’ BM, although this is not shown in the present data and must be investigated further in independent and larger pediatric AML cohorts.

## 4. Discussion

In this explorative study, we presented leukemia-associated protein profiles in stem cells and blast subsets from children with AML and compared them with HSCs/HPCs from age-matched hematologically healthy controls. Interestingly, based on the protein expression patterns, PCA analyses showed that the AML-SCs from pAML23 and pAML29 samples clustered with their CD34+ and CD34− blasts and in a different position than the healthy cell subsets. By contrast, the AML-SCs from pAML17, pAML20, and pAML21 displayed protein profiles that were, in part, similar to the control samples. Of particular interest was that the AML-SCs from pAML17 and pAML21 were positioned near the HSCs, suggestive of a more HSC-like protein profile in these AML-SCs. Still, the present study is a hypothesis-generating study and to obtain biological proof of these profiles and their putative functional impact, the data must be investigated in independent and ideally larger pediatric AML cohorts.

Using IPA, we explored whether the proteins that were differentially expressed between the AML-SCs and HSCs were suggestive of pathway perturbations that could be of potential leukemogenic importance. Here, we found that signaling through the EIF2 pathway was significantly downregulated in AML-SCs. Physiologically, EIF2 signaling is important for protein translation initiation, which is inhibited in relation to cellular stress, resulting in reduced global protein synthesis [43]. Recently, Van Galen et al. reported that EIF2 signaling was downregulated in cord blood HSCs compared to myeloid progenitors as part of the so-called integrated stress response, and acted to conserve cellular homeostasis [44]. Additionally, the authors demonstrated that EIF2 signaling was downregulated in leukemic stem cells compared with leukemic blasts from adult AML samples; however, no direct comparison between HSCs and leukemic stem cells was performed [44]. Collectively, our results and the data from Van Galen and colleagues imply that inhibition of EIF2 signaling plays a central role in AML that awaits further investigation.

We explored the protein profiles discriminating the AML-SCs, PC1s, and BC1s from HSCs. Overexpression of NUP210 was found in all analyses. In the context of both healthy hematopoiesis and leukemia, the function of this particular nucleoporin remains to be described. However, other nucleoporins, such as NUP98 and NUP214, are recurrently involved in gene rearrangements in AML [45], making further explorations of the nucleoporins appealing. In addition, HSPE1, a member of the heat shock family of proteins acting as an HSP60 co-chaperone [46], and SRSF1, a splicing regulator [47], were upregulated in AML-SC. Both proteins have previously been shown to be upregulated in several cancers, including AML and pediatric acute lymphoblastic leukemia [46,47,48,49]. Interestingly, increased levels of CALR were identified in the leukemic blast compartments. In myeloproliferative neoplasms, CALR gene mutations are recurrent events [50], and it is well-known that stress-induced CALR exposure on the cell surface is considered a prophagocytic signal mediating immunogenic cell death [51]. The overexpression of surface CALR has been demonstrated in several types of cancer cells and also on leukemic blasts and AML-SCs in adult AML compared with healthy control HSCs [52,53]. In the present study, the nature of the proteomic approach used did not allow us to determine the cellular localization of CALR. Additionally, in contrast to the aforementioned data, we did not identify CALR overexpression in AML-SCs when compared with HSCs. However, a correlation between high CALR surface exposure and reduced EIF2 signaling has previously been shown [53,54]. This adds to the proposition that the EIF2 signaling pathway and the interplay between EIF2 and CALR may be important in the pathogenesis of AML. However, these findings need verification and validation before any potential implication in leukemogenesis or prognostic impact can be inferred.

Our data show that proteomic profiling of highly purified cell populations is technically feasible even with numbers as low as 1500 purified cells. A previous study, using reverse phase protein array (RPPA) of 121 selected protein targets, identified differences in the protein profile between AML-SCs and the bulk leukemia and leukemic CD34+ cells from adult AML samples [55]. Due to too few HSCs for separate analyses, the authors compared AML-SCs with healthy CD34+ BM cells [55]. However, as our data indicated, the protein profiles of the various cell stages of healthy hematopoiesis differed. Therefore, proteomic analysis on individual subsets is needed to identify the specific protein expression patterns separating AML-SC from HSCs, which will unquestionably be missed when compiling immature hematopoietic subsets into one. In fact, recent studies comparing the proteome of AML-SCs from six adult AML patients with their corresponding blasts and with healthy adult HSCs described specific pathways enriched within the AML-SC pool, such as degradation of branched-chain amino acids (BCAA), oxidative phosphorylation, and spliceosome pathways [29,56].

One major strength of our data is that we de facto compared pediatric AML samples with the most biologically relevant control group of hematologically healthy, age-matched controls because age-related biological differences in HSCs have been established [27]. Because children need to be under general anesthesia before BM sampling, this could only be performed in parallel with planned procedures such as surgery of the hip/pelvic region, which made it impossible to obtain a larger control group.

Limited literature exists on proteomic profiling of pediatric AML. Prior studies have employed RPPA and 2D gel electrophoresis together with matrix-assisted laser desorption ionization time-of-flight MS for proteomic analysis of unfractionated BM samples [57,58,59], and, recently, Nguyen et al. compared the global proteomic profile of unfractionated leukemic bone marrow samples from 16 pediatric AML patients with or without core binding factor AML [60]. To the best of our knowledge, the use of highly purified cell populations instead of unfractionated BM samples has not previously been documented in pediatric AML. This strategy reinforced the specific cell population signals in both leukemic and healthy cell subsets. It is a challenge and a compromising balance to reduce cellular heterogeneity by isolating immunophenotypically identical cell populations while obtaining enough cellular input for the MS analysis. We analyzed samples with down to 1500 cells, resulting in a limited number of proteins for downstream analyses, as well as only allowing for analyses of proteins identified in all samples. Together with the high degree of both genotypic and phenotypic heterogeneity of the limited number of patients included, this complicated the interpretation of the results and prevented the establishment of final conclusions related to fundamental AML-SC biology in pediatric AML. Nevertheless, our results are explorative and hypothesis-generating, and should be further explored in larger cohorts of pediatric AML to establish any possible leukemogenic impact of the identified differential expressed proteins and their potential as future treatment targets. Moreover, since we did not perform correction for multiple hypothesis testing in order not to increase type 2 errors with the risk of overlooking putative markers, it is further warranted that the results are verified in other independent pediatric AML cohorts.

## 5. Conclusions

In conclusion, we demonstrate altered protein expression patterns in purified leukemic cell subsets in pediatric AML. Importantly, the comparison of AML subsets to normal stem and progenitor cells is a way of obtaining a specific insight into the altered biological processes of potential therapeutic interest. In theory, severe toxicities such as myeloablation might be avoided by targeting proteins or pathways that spare, at least in part, HSCs. Our data pave the way for future investigations and the identification of such perturbed pathways that may when further investigated provide information on presumed molecular pathogenetic mechanisms and/or give rise to potential new candidates for targeted therapy in children with AML.

## Figures and Tables

**Figure 1 cancers-14-03567-f001:**
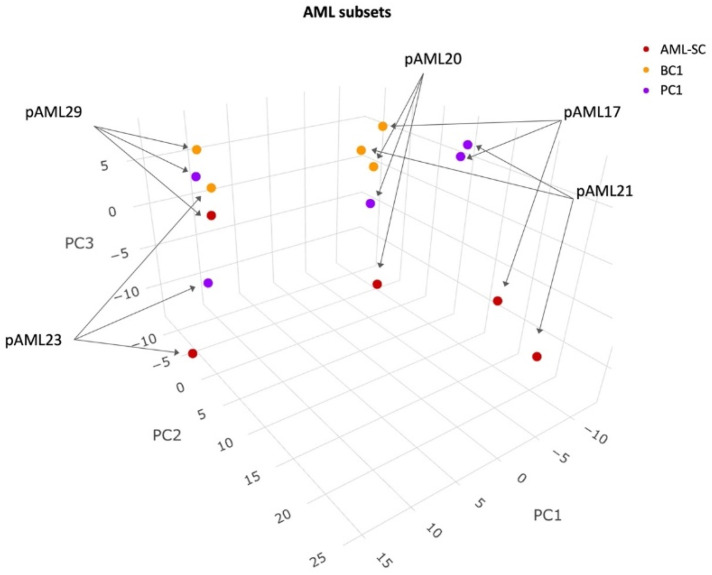
PCA of leukemic cell subset protein expression pattern. Color-coding and sample ID are indicated. Samples from pAML23 and pAML29 occupied a separate space in the map. The samples from pAML17, pAML20, and pAML21 were more widely scattered, although the BC1 samples clustered.

**Figure 2 cancers-14-03567-f002:**
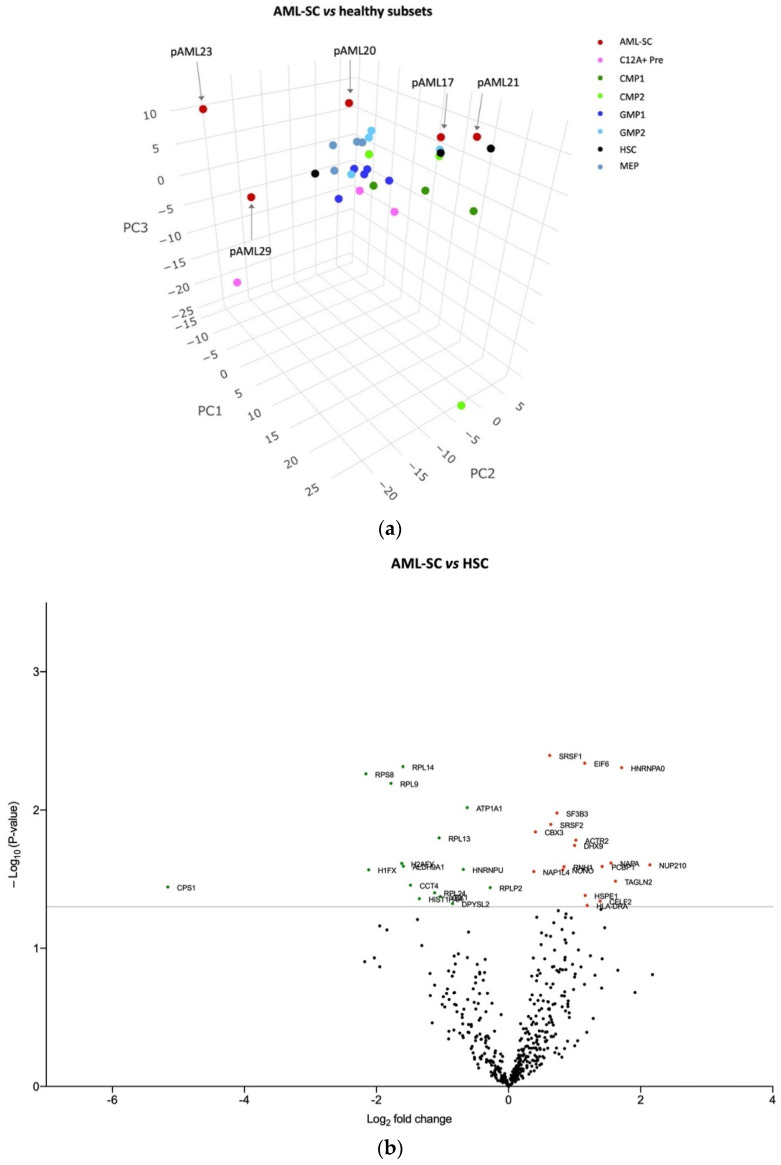
The protein profile differs in AML-SCs compared with HSCs and HPCs. (**a**) PCA of AML-SC and healthy immature cell subset protein expression pattern. Color-coding and the sample ID of the AML-SCs are indicated. The AML-SCs from pAML17 and pAML21 were situated in close relation to healthy HSCs as opposed to AML-SCs from pAML20, pAML23, and pAML29; the latter two did not map near the healthy counterparts. (**b**) Volcano plot displaying proteins variably expressed between AML-SCs and HSCs. In all, 34 proteins were differentially expressed (*p*-value < 0.05). Upregulated proteins: red spheres. Downregulated proteins: green spheres. The gray lines indicate a *p*-value of < 0.05. The labeled proteins are listed in Appendix A. (**c**) Ingenuity pathway analysis of the 34 proteins differentially expressed in AML-SCs. The observed changes in protein expression levels are indicated in red (upregulated) and green (downregulated) shown in the periphery. Upstream analysis in IPA revealed seven putative regulators that could explain this (inner circle of proteins). Six of the seven proteins are transcription factors, while one (SFPQ) is a splicing factor, all participating in gene expression. Decreased activity of one of the transcription factors (MLXIPL, blue) could explain the significant downregulation of the “signaling through EIF2” pathway identified in the IPA analysis. Stippled lines: indirect relationship. Solid line: direct relationship [42].

**Figure 3 cancers-14-03567-f003:**
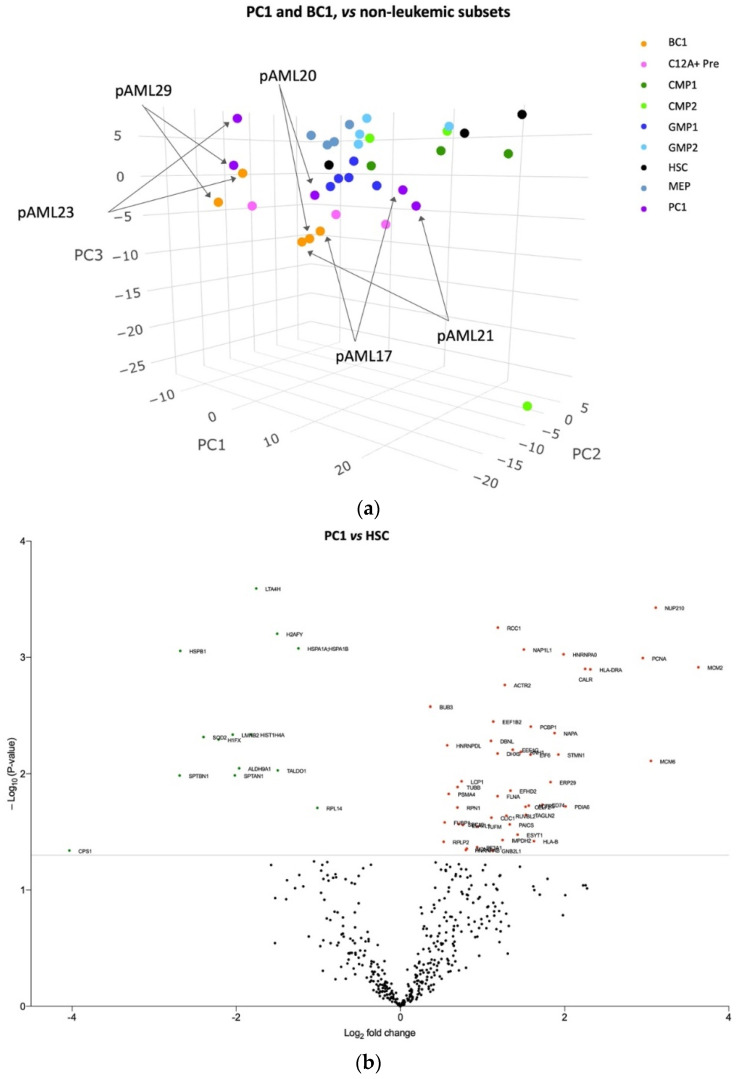
Differences in protein profiles between leukemic blasts and healthy HSCs and HPCs. (**a**) PCA of leukemic blast and healthy immature cell subset protein expression pattern. Color-coding and the sample ID of the leukemic samples are indicated. The leukemic blasts from pAML23 and pAML29 were situated separated from the other leukemic and healthy subsets. The leukemic blasts from pAML17, pAML20, and pAML21 were positioned parallel to the healthy progenitor subsets. (**b**) Volcano plot displaying differentially abundant proteins between PC1 and HSCs. A total of 62 proteins were differentially expressed (*p*-value < 0.05). Upregulated: red spheres. Downregulated: green spheres. The labeled proteins are listed in Appendix A. (**c**) Volcano plot displaying differentially abundant proteins between BC1 and HSCs. A total of 132 proteins of differential abundance were identified (*p*-value < 0.05) Upregulated: red spheres. Downregulated: green spheres. The labeled proteins are listed in Appendix A.

**Table 1 cancers-14-03567-t001:** Patient characteristics.

Sample ID	Age	Sex	FAB Type	CD34 Category (%) ^#^	Karyotype	Known Mutations	CD34+CD38– Mutational Status	Events
pAML17	2	female	M5	CD34 negative (0.07%)	46, XX, inv(6)(p12q16), t(9;11)(p22;q23) [25]	none	t(9;11) not detected	Induction death
pAML20	12	male	M4	CD34 positive (8.6%)	46, XY	none	unknown	Resistant disease to death
pAML21	4	female	M5	CD34 negative (0.12%)	46, XX, t(9;11)(p22;q23) [16]/47, idem, +9 [9]	none	t(9;11) not detected	Death after relapse
pAML23	11	female	M4	CD34 positive (71.5%)	46, XX	none	unknown	Induction death
pAML29	7	female	M5	CD34 positive (37.4%)	46, XX, inv(16)(p13q22) [25]	*FLT3*-TKD	Inv(16) *FLT3*-TKD	Induction death

*FLT3*-TKD: FLT3 tyrosine kinase domain; ^#^ Patient samples were categorized as CD34 positive when > 5% of the myeloid blasts were CD34 positive. The fraction of CD34+ cells for each patient is shown in parentheses.

## Data Availability

These data analyzed during the current study are available upon reasonable request.

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
