# Peer review of "Proteomic Profiling Identifies Specific Leukemic Stem Cell-Associated Protein Expression Patterns in Pediatric AML Patients"

_cancers, 2022, doi:10.3390/cancers14153567_

Round 1

Reviewer 1 Report

The authors analyzed the proteomic profile of leukemic blasts and acute myeloid leukemia stem cells and compared the differences with the one of normal hematopoietic  stem cells. Several proteins were found differentially expressed with some of them being up-regulated (NUP210, SRSF1, HSPE1). Two proteins (NUP210 and Calreticulin) were found up-regulated in blasts defined according to CLEC12A positivity.

The study is very interesting. The approach is very valid and the analysis has been conducted in primary specimens of pediatric AML patients and healthy subjects. Therefore I am very enthusiastic about it. The manuscript is presented well in the text. However I have some comments to improve the manuscript.

1. Figures are extremely small. Resolution should be improved for all. 

2. The proteomic analysis should be combined with additional techniques to strengthen the data. Flow cytometry or western blotting should be attempted on the specimens (otherwise the analysis is only speculative). 

3. The authors discussed the unbalance between protein expression and mRNA expression pointing out the lack of correlation. I believe that this is true but cannot be true for all proteins and genes observed. The authors should do analysis on publicly available data to add more evidence and report the figures in the revised manuscript. 

4. The discussion is very long. It needs to be shortened. Also please remove sentences at lines 275-278. They represent a comment from a reviewer. 

Author Response

Reviewer #1

The authors analyzed the proteomic profile of leukemic blasts and acute myeloid leukemia stem cells and compared the differences with the one of normal hematopoietic  stem cells. Several proteins were found differentially expressed with some of them being up-regulated (NUP210, SRSF1, HSPE1). Two proteins (NUP210 and Calreticulin) were found up-regulated in blasts defined according to CLEC12A positivity.

The study is very interesting. The approach is very valid and the analysis has been conducted in primary specimens of pediatric AML patients and healthy subjects. Therefore I am very enthusiastic about it. The manuscript is presented well in the text. However I have some comments to improve the manuscript.

  1. Figures are extremely small. Resolution should be improved for all. 

Response: All figures have initially been submitted in high resolution in separate files. However, the reviewers have presumably only access to the figures in the word document. If the initially submitted high resolution figures does not match the requirement of the journal, please do not hesitate to contact us.

  1. The proteomic analysis should be combined with additional techniques to strengthen the data. Flow cytometry or western blotting should be attempted on the specimens (otherwise the analysis is only speculative). 

Response: These samples collected for pediatric patients with AML are very limited due to the rarity of the disease and we unfortunately do not have more sample material from the patients included to analyse the specified subpopulations by WB or FCM. In the present study, our primary focus was to obtain as pure FACS sorted cell populations as possible and thus with the very limited cell numbers, all sample material were used for the proteomic analysis. The NOPHO (Nordic Society of Pediatric Haematology and Oncology) consortium includes biobanking of samples from the pediatric patients and is the largest of its kind in the world – and even with this consortium only bone marrow samples archived as live cells in DMSO in liquid nitrogen from few children with AML exists. Furthermore, as we have found that the group of pediatric AMLs are truly heterogeneous, future studies must be focused on either CD34+ or CD34- pediatric AML patients. Also, up front we did perform flow cytometric, FACS and genetic analyses of stem and blast cell subsets from three (pAML17, pAML21 and pAML29), from which we had extra material, as well as in the five pediatric AML sample harboring genetic abnormalities at time of diagnosis. This information is mentioned in lines 107–109 and in the result section lines 162-173 with the reference to our initial study on immunophenotyping LSCs in these patients. Maybe the reviewer did not have access to our supplemental date in which this is describes. The results are shown in Table 1.

  1. The authors discussed the unbalance between protein expression and mRNA expression pointing out the lack of correlation. I believe that this is true but cannot be true for all proteins and genes observed. The authors should do analysis on publicly available data to add more evidence and report the figures in the revised manuscript. 

Response: We thank the reviewer for this comment. Paired proteomic and transcriptomic data from the isolated cell populations would have been highly interesting to compare. However, as the available biological material was sparse, all material was used for the proteomic analyses. Also, as some of the cell populations were rare – for example the stem cells – we were only able to FACS sort a limited number of cells. Technically, we do not have the set up to perform gene expression profiling on as little as 1500 cells. To clarify that we do not have transcriptomic data, the section discussing the comparison of proteomics and mRNA have been removed from the discussion (see below). Furthermore, to the best of our knowledge no publicly available transcription dataset nor published manuscripts with included transcription expression data exist on the FACS sorted populations in pediatric AML we present in the current study. To investigate the mRNA/protein expression profiles for example with previously published datasets (on other pediatric AML patients), it is imperative that this is performed on identical subpopulations in order to analyze the relation between the mRNA levels and the protein expression levels. Currently, this is unfortunately not conceivable.

  1. The discussion is very long. It needs to be shortened. Also please remove sentences at lines 275-278. They represent a comment from a reviewer. 

Response: We have removed the sections lines 336-340 and lines 362-367 to shorten the discussion and to remove the speculative sections on RNA expression. However, in addition only few changes have been made to the discussion, mostly to clarify when our statements are speculative and not based on the current data results.

Regarding the lines 275 – 278, (now 293 - 296): Thank you for noticing this remaining sentence from the initial document from Cancers. This now removed.

Reviewer 2 Report

The paper by Petersen et al. describes a mass-spectrometry-based profiling of the proteomes of leukemic blasts and stem cells in five pediatric acute myeloid leukemia compared with five normal hematopoietic stem cells. The authors identify differentially expressed proteins, and they discuss their potential interest as targets for AML treatment.

While the subject is clinically relevant, the study is a mere description of a single proteomics experiment, performed on a limited number of patients. The quality of the display of the results is very poor, and there is no validation whatsoever of the potential targets identified. Therefore, I believe that this manuscript is too preliminary and not suitable for publication in Cancers, and may find a better match in a more specialized proteomics journal, after improving the description and the display of the results.

More specific comments:

- the volcano plots reported in figures 2 and 3, which report the main findings of this paper, are absolutely illegible.

- the methods concerning the database search and data analysis contain far too few details and are inadequate. Missing information includes (but is not limited to) the software and the detailed parameters used for the database search, the filtering criteria, and the parameters used for the analysis and display of the data.

- a simple Student’s t-test is not the most appropriate test when dealing with a high number of comparisons. A test corrected for multiple comparisons should be used.

Author Response

Reviewer #2

The paper by Petersen et al. describes a mass-spectrometry-based profiling of the proteomes of leukemic blasts and stem cells in five pediatric acute myeloid leukemia compared with five normal hematopoietic stem cells. The authors identify differentially expressed proteins, and they discuss their potential interest as targets for AML treatment. 

While the subject is clinically relevant, the study is a mere description of a single proteomics experiment, performed on a limited number of patients. The quality of the display of the results is very poor, and there is no validation whatsoever of the potential targets identified. Therefore, I believe that this manuscript is too preliminary and not suitable for publication in Cancers, and may find a better match in a more specialized proteomics journal, after improving the description and the display of the results.

Response: We agree with the reviewer that the present study is merely a hypothesis-generating study and from the data presented, we are not able to delineate the biology behind pediatric AML. However, based on the severity and rarity of the malignancy, we still find the study to be justified and being the first of its kind providing a hypothesis-generating proteomic landscape in FACS sorted AML stem cells and HSCs in pediatric AML. The results will hopefully be validated but this will require other independent cohorts of pediatric patients with AML. We have throughout the manuscript underlined that the data needs further investigation prior to be able to determine the functional/biological/clinical impact. In addition, we have gone through the manuscript and rewritten/added paragraphs where we initially may have been too hypothetical about our data, i.e., lines 98-99, 289-291, and 305-308.

Unfortunately, the reviewers may not have had access to our high-resolution figures in the review process, thus making the evaluation of the figures impossible. We hope that these will be available for the second review

More specific comments:

- the volcano plots reported in figures 2 and 3, which report the main findings of this paper, are absolutely illegible.

Response: All figures have initially been submitted in high resolution in separate files. However, the reviewers have presumably only access to the figures in the word document. If the initially submitted high resolution figures does not match the requirement of the journal, please do not hesitate to contact us.

- the methods concerning the database search and data analysis contain far too few details and are inadequate. Missing information includes (but is not limited to) the software and the detailed parameters used for the database search, the filtering criteria, and the parameters used for the analysis and display of the data.

Response: All this is described in Supplementary Methods in the Supplementary Materials document. Unfortunately, the reviewers may not have had access to our Supplementary document in the review process, thus this comment on missing information. We hope that this will be available for the second review

- a simple Student’s t-test is not the most appropriate test when dealing with a high number of comparisons. A test corrected for multiple comparisons should be used.

Response: We agree with the reviewer that ideally a correction for multiple hypothesis testing, decreasing the type 1 error, would be required in order to make more solid conclusions. However, we performed a discovery-based, hypothesis generating study on a very limited number of clinical samples from children with a very rare disease. In that light we find that a major disadvantage would be the increased type 2 error with the risk of overlooking novel, putative markers. We realize that the data of cause need to be verified, preferably on a different and larger cohort. We have included these thoughts in the manuscript, lines 153-156, and 393-396

Round 2

Reviewer 1 Report

The authors answered appropriately to my queries. No further comment.

Reviewer 2 Report

While the authors addressed two of the comments that I made (but the volcano plots are still illegible, and I do not believe it is a matter of resolution, but rather of the size of the labels), I believe that the results presented in this study – in the absence of any validation/experiments performed with alternative techniques – are too preliminary for the Cancers journal. I do understand the difficulty in finding the samples and the scarcity of the starting material, but reporting a single proteomics experiment is simply not enough, in my opinion. As I mentioned previously, this manuscript may be better suited for a proteomics journal.